# FRP-Reinforced/Strengthened Concrete: State-of-the-Art Review on Durability and Mechanical Effects

**DOI:** 10.3390/ma16051990

**Published:** 2023-02-28

**Authors:** Jesús D. Ortiz, Seyed Saman Khedmatgozar Dolati, Pranit Malla, Antonio Nanni, Armin Mehrabi

**Affiliations:** 1Department of Civil and Architectural Engineering, University of Miami, Coral Gables, FL 33146, USA; 2Department of Civil and Environmental Engineering, Florida International University, Miami, FL 33174, USA

**Keywords:** FRP composites, thermoset composites, GFRP bars, CFRP fabrics, durability, serviceability, mechanical properties, environmental effects

## Abstract

Fiber-reinforced polymer (FRP) composites have gained increasing recognition and application in the field of civil engineering in recent decades due to their notable mechanical properties and chemical resistance. However, FRP composites may also be affected by harsh environmental conditions (e.g., water, alkaline solutions, saline solutions, elevated temperature) and exhibit mechanical phenomena (e.g., creep rupture, fatigue, shrinkage) that could affect the performance of the FRP reinforced/strengthened concrete (FRP-RSC) elements. This paper presents the current state-of-the-art on the key environmental and mechanical conditions affecting the durability and mechanical properties of the main FRP composites used in reinforced concrete (RC) structures (i.e., Glass/vinyl-ester FRP bars and Carbon/epoxy FRP fabrics for internal and external application, respectively). The most likely sources and their effects on the physical/mechanical properties of FRP composites are highlighted herein. In general, no more than 20% tensile strength was reported in the literature for the different exposures without combined effects. Additionally, some provisions for the serviceability design of FRP-RSC elements (e.g., environmental factors, creep reduction factor) are examined and commented upon to understand the implications of the durability and mechanical properties. Furthermore, the differences in serviceability criteria for FRP and steel RC elements are highlighted. Through familiarity with their behavior and effects on enhancing the long-term performance of RSC elements, it is expected that the results of this study will help in the proper use of FRP materials for concrete structures.

## 1. Introduction

Corrosion is a major problem for steel-reinforced concrete (RC) structures, being responsible for the deterioration of the physical-mechanical properties of the rebar (e.g., the reduction of the steel cross-section), the rebar to concrete interface (i.e., loss of the bond between the steel and concrete), and the concrete cover (cracking from expansive corrosion) [1,2]. There are several alternatives to prevent or mitigate corrosion (i.e., coating techniques on steel, cathodic protection, corrosion inhibitors, stainless steel rebars [3,4,5]), and techniques to strengthen, retrofit, and repair deteriorated steel-RC structures (e.g., external plate bonding, section enlargement, external post-tensioning, and the injection of epoxy [6,7]). Fiber-reinforced polymer (FRP) composites have in recent decades emerged as a suitable alternative to steel due to their notable mechanical properties and chemical resistance. They are known for having high longitudinal tensile strength, no corrosion, light weight, and anisotropic and elastic behavior up to rupture [8,9,10,11].

FRP composites consist of reinforcing fibers embedded in a polymer matrix. Continuous glass, carbon, basalt, and aramid fibers are the types of fibers in composites that are used for structural engineering applications [12], and thermoplastic and thermosetting are the two polymer resins, with the latter being most commonly employed. Moreover, different additives and fillers are typically mixed with the polymer resin as well as proprietary sizings added to the fibers (i.e., thin coating applied to the surface of the fiber) to improve wettability and fabrication, tailor the composite performance, and reduce costs [11,12]. This way, the FRP composites can be employed for both internal and external applications in RC structures.

Despite their favorable properties, FRPs may be subject to fatigue and creep rupture which, together with their low modulus of elasticity (compared to steel) make the serviceability limit states typically control the flexural design of concrete members reinforced with GFRP (Glass/vinyl-ester FRP) bars that exhibit lower stiffness [13]. Additionally, harsh environmental conditions (i.e., the presence of water, alkaline or acid solutions, saline solutions, elevated temperature, and ultraviolet light exposure) could also affect the strength and stiffness of FRPs [14]. Available design codes and guides in North America and Japan [15,16,17,18,19,20] provide strength reduction and environmental factors to account for these conditions. Furthermore, as the number of FRP-RSC structures increases, so does the need to perform inspections on them to assess the current condition of their elements. However, there are currently no provisions for such a task as there are for steel structures [21].

The present paper reviews the current state-of-the-art on the main environmental and mechanical conditions affecting the mechanical properties of FRP composites and their impact in the design requirements for RSC elements. First, a brief background on FRP composites and their use as reinforcement and strengthening materials is presented, followed by a classification and presentation of the factors leading to the deterioration of FRP composites. Chapter 3 examines the main environmental effects on FRP composites, including exposure to water, saline, alkaline, UV, freeze-thaw, and elevated temperatures exposure. Chapter 4 focuses on the main mechanical effects affecting the mechanical properties of FRP composites and FRP-RSC elements, including fatigue, creep rupture, and shrinkage. In Chapter 5, the impact on design requirements of GFRP-RSC structures is discussed. Finally, the paper concludes with a summary and conclusions.

To narrow the wide range of composite materials that are available, this review only focuses on Glass/vinyl-ester FRP bars for internal application and Carbon/epoxy FRP fabrics for external application, which are the most commonly studied in the literature and present in the US market. In addition, for internal application, the new ACI 440.11 code [15] references ASTM D7957 [22], which only allows the use of this thermoset composite (i.e., vinyl-ester) for the production of GFRP bars. Although the traditional reviews differentiate between FRP reinforcement and FRP strengthening applications, the environmental factors affect the composite in similar ways for both applications, differing only in the level of exposure. It should also be noted that the future scenarios may include the strengthening of structures that are originally reinforced with FRP bars, which is another reason to consider the external and internal use of FRP together. This serves as a starting point for the recognition of serviceability issues of FRP-RSC during inspection that may be different from those of steel RC.

## 2. Background on FRP Composites

The development of FRP composites began in the 1930s when the first patented fiberglass was produced by Owens Glass company, who joined with the Corning company [23]. Later, during World War II, the FRP industry made significant progress that continued during the next twenty years, with applications mostly in the military and aerospace industries [24]. Numerous research studies have been conducted since then to study the potential applications of FRP composites in structural engineering. FRP applications in concrete structures can be divided into two categories: internal applications with FRP bars, rods, and tendons, and external application with FRP plates, fabrics, wraps and near-surface mounted (NSM) FRP bars. The former also commonly refers to new construction (i.e., FRP-reinforced concrete structures and FRP-prestressed concrete structures), while the latter is associated with the strengthening, retrofitting, and repair of existing structures, typically steel-reinforced concrete structures, but in the future FRP-reinforced concrete structures may require strengthening. A tree diagram of FRP composites, including its components and applications, is shown in Figure 1.

### 2.1. Internal Application

The internal application of FRP composites is strongly related to locations where the corrosion of traditional steel reinforcement is an economic and safety concern [25]. The good mechanical performance, exceptional durability, and sustainability implications (i.e., lower environmental impacts such as global warming, photochemical oxidant creation and acidification, compared to steel-RC [26]) of the FRP composites [7,26,27,28] make them suitable for these conditions. Additionally, when compared to conventional steel rebars, FRP bars have significantly higher strength (e.g., GFRP bars could have an ultimate tensile strength ranging between 550 MPa (80 ksi) and 1380 MPa (200 ksi) [29]), about one-fourth of the density of steel (reducing the overall weight of the structure and requiring less scaffolding and heavy equipment), and can achieve a longer service life [4]. However, the difference between their coefficients of thermal expansion (the longitudinal one controlled by the fibers and the transverse one by the resin), and challenges related to its vulnerability to elevated temperatures and lack of bendability (specifically for thermoset resins, thermoplastics has been proven to allow bending FRP composites [30,31]) are listed as some drawbacks that limit the use of the FRP composites in specific situations [32]. Although its initial cost is also frequently highlighted as one the main drawbacks to its implementation (mainly when compared to the price of steel rebars), the initial cost of GFRP rebars has changed significantly over the past two years due to price fluctuations in the metal market worldwide since the mid-2020s and the growth of the GFRP rebar market [33]. The cost for a #8 GFRP straight bar delivered on the East Coast of the USA is estimated to be between US$1.70 and US$1.80 per linear foot compared to around US$1.50 per linear foot of black steel (based on US$0.55 per pound) and US$2.14 per linear foot of epoxy-coated steel rebars (based on $0.80 USD per pound) [34] (These figures are representative of retail prices and do not account for other factors such as transportation or discounts for order size). Furthermore, the long-term cost of FRP reinforced concrete structures can be lower compared to traditional reinforcement due to their corrosion resistance and durability, which can result in reduced maintenance and repair costs over the structure’s lifetime [26].

Despite some challenges, new construction projects using FRP as the primary or auxiliary reinforcement in concrete structures have been developed all over the world [12,35,36,37,38,39]. In recent years, the increasing collaborative research between academia, industry, and organizations has yielded important results that have provided updated material standards [40] and design specifications [15,17,18,20,41,42], which owners, engineers, and contractors will have at their disposal for the safe construction and inspection of FRP-reinforced concrete structures [40] (see Table 1) Several organizations have developed guidelines for the design of reinforced concrete (RC) structures with FRP composites. Some of these worldwide organizations include the American Association of State Highway and Transportation (AASHTO), the American Concrete Institute (ACI), the Canadian Standards Association (CSA), and the Japan Prestressed Concrete Institute (JPCI). For internal applications, the available design guidelines/specifications are usually focused on GFRP bars or CFRP strands. The new ACI 440.11 code [15] is one of the most significant improvements in the last few years in providing minimum requirements for materials, design, and construction of GFRP-RC structures in a mandatory language. However, more work needs to be done before this technology can be fully implemented [40].

### 2.2. External Application

The external application of FRP composites has primarily been implemented for the rehabilitation of structures when design defects are noticed, an in-service element’s load-bearing capacity is modified, the materials have begun to deteriorate, or extreme events have occurred (i.e., seismic or fire events) [43]. In addition to the previously mentioned advantages of FRP materials, their strength-to-weight ratio, ease of installation, and suitability for irregular surfaces are additional reasons that have contributed to their popularity as a strengthening technique [44].

External FRP systems are generally applied to RC elements to enhance the flexural, axial, shear, and torsional strength, and, in seismic zones, FRP wraps can be used for columns to increase ductility due to the induced confinement of the concrete [7,44,45,46]. FRP laminates (wraps/fabrics, strips, and plates) are the most common externally bonded FRP systems for strengthening existing structures. The application techniques can be categorized into “wet layup,” “prepreg,” and “precured” systems, which differ in the time of application of the polymer resin [47]. Commonly, the same polymer resin (occasionally with some additives) is also employed to act as an adhesive in plates or as a primer, putty coat, and saturant in wet lay-up systems [48].

Experimental studies have shown different failure modes in elements strengthened with FRP fabrics. The most prevalent failure modes, as per ACI 440.2R-17 [16], include FRP fabric rupture, the debonding of FRP fabrics from the concrete surface, and concrete cover delamination [16]. Because of the complexity of the different failure modes, adequate material standards, design specifications, installation procedures, and inspection guidelines are essential to achieving full acceptance of the FRP composites in the structural engineering community. Four available standards and specifications for the design of reinforced/strengthened concrete structures are shown in Table 1. Nevertheless, no inspection guidelines have been developed to address maintenance for structures exposure to harsh environments or suffering from potential defects in FRP. These defects in FRP applications can be attributed to a variety of reasons, including mechanical, environmental and design factors, fabrication, and workmanship.

**Table 1 materials-16-01990-t001:** Available design guides and codes focused on FRP composites.

Application	Internal Application	External Application	Material Specifications
AASHTO	LRFD Bridge Design Guide Specifications for GFRP-Reinforced Concrete (2nd ed.) [18]	Guide Specifications for Design of Bonded FRP Systems for Repair and Strengthening of Concrete Bridge Elements (1st ed.) [19]	*ASTM International* D7205/D7205M-06(2016). D7914/D7914M. D7957/D7957M-22
ACI	ACI 440-22. Code Requirements for Structural Concrete Reinforced with Glass FRP Bars (1st ed.) [15]	ACI 440.2R-17. Guide for the Design and Construction of Externally Bonded FRP Systems for Strengthening Concrete Structures [16]	ACI SPEC-440.5-22. Construction with Glass Fiber-Reinforced Polymer Reinforcing Bars [49]
CSA	CSA S806-12 (R2017). Design and construction of building structures with fibre-reinforced polymers [20]	CSA S806-12 (R2017). Design and construction of building structures with fibre-reinforced polymers [20]	CSA S807-19. Specification for fibre-reinforced polymers [50]
JPCI	Recommendation for Design and Construction of Concrete Structures using Fiber Reinforced Polymers (FRP)	-	-

The mechanical and environmental effects on FRP composites have been extensively discussed in the available literature (their performance under long-term effects still needs to be further addressed to ensure an adequate understanding [7,9]). However, due to the broad spectrum of FRP composite types and because, as previously described, the internal and external application in reinforced concrete (RC) is primarily focused on the use of Glass FRP bars (ACI 440.11-22 only addresses design with Glass/vinyl-ester FRP rebars) and Carbon FRP fabrics, respectively. The following discussion is centered on these types of FRP composites and their performance when used in internal and external applications.

## 3. Environmental Effects

Physical-mechanical properties of the FRP composites, as well as the concrete surrounding or supporting them, could be affected when they are subjected to harsh environmental conditions. Nevertheless, due to the variety in fibers (i.e., glass, carbon, aramid, or basalt); polymer resins (e.g., polyester, epoxy, vinyl ester, etc.); fiber volume fractions; solution concentrations; temperature; among others, it is not proper to stipulate a single range of deterioration, even for a specific composite (e.g., E-glass/vinyl ester) subjected to a specific harsh environment [9]. Current research on the environmental effects on FRP bars focuses mainly on GFRP and BFRP bars [14] for internal applications and CFRP laminates for external applications where harsh environmental conditions (i.e., high humidity, seawater and deicing salt ambient, acid rain, and pore solution of concrete) are simulated to study their impact.

### 3.1. Water Exposure

The fibers, matrix, and fiber-matrix interface of an FRP composite can degrade when exposed to water, high moisture, or high humidity. The degradation mechanisms of Glass FRP composites under water exposure include the entry of moisture through the fiber-matrix interface [51], leading to the progression of microcracks, which accelerates the diffusion of water and chemicals [52], and the plasticization and hydrolysis impacting the polymer resin; the former softens the resin and hence reduces the stiffness of the FRP composite, whereas the latter irreversibly breaks and weakens the bonds in the polymer chains [53,54].

GFRP bars

Kim et al. [55] stated that the tensile strength retention of E-glass/vinyl ester bars (73% of fiber content by weight) was about 88% and 80% after 132 days immersed in tap water at room temperature and 40 °C, respectively. The elastic modulus of elasticity was reduced by up to 25%, and the interfacial shear strength loss (to evaluate the degradation of interface between fiber and matrix) was about 20%. Al-Salloum et al. [56] noted an improvement on the new generations of GFRP/vinyl ester bars (83% of fiber content, although not specified, it is assumed to be by weight) by observing better residual tensile strength on the tested bars compared to most of the literature at the time. All bars presented a brittle fracture with delamination as the mode of failure. The presence of elevated temperature impacts the diffusivity, which accelerates the tensile strength deterioration by up to 34% at 80 °C [55]. Figure 2 shows the tensile strength retention versus the exposure period for GFRP bars immersed in water solution from the available literature [55,56,57,58,59]. However, due to the lack of enough data to analyze the differences in the results from different parameters such as fiber type, fiber volume fraction, polymeric matrix, bar diameter, and bar surface treatment that could impact the performance, it is not possible to establish accurate prediction models [9,14]. D’Antino & Pisani [60] analyzed fifty-seven tensile tests of bars immersed in water solutions, and found an average residual strength ratio of 93.5%, 87.3% and 67.2% for temperatures between 11–25 °C, 26–50 °C and 60–80 °C, respectively. Al-Salloum et al. [56] and D’Antino & Pisani [60] can be consulted for a more comprehensive analysis of the impact of water exposure on GFRP bars.

Furthermore, as GFRP bars are embedded in concrete for internal application during their service life, Nepomuceno et al. [61] reviewed the long-term bond behavior of FRP bars to concrete based on pull-out tests, and concluded that the exposure environments, exposure times (up to 5760 h), and exposure temperatures (20 °C and 80 °C) barely affect the bond behavior, finding very high strength retention rates, even ones close to 100%.

CFRP fabrics

In FRP fabric systems (sheets), the polymer resin (usually epoxy resin) is utilized as a primer, putty coat, and saturant in externally bonded applications. To serve as an adhesive between the FRP system and the concrete surface, the resin is primed onto the surface of the element. Therefore, the presence of humidity or water during installation or the service life of the element could be detrimental to epoxy resin due to its water absorption capacity [48,62,63]. The epoxy resin is also commonly used to bond the FRP plies together when more than one layer is used. This interface between the adjacent layers is more prone to water-induced damage [63,64]. When combined with high temperature (i.e., 38 °C), the hygrothermal environment affect the color of the composite (which could change from dark to light grey) [65]. A study on CFRP externally bonded systems (both wet-layout and laminates) observed that the flexural strength decreased rapidly in the first 50 days and slowly afterwards regardless of water temperature [66]. The failure mode of the strengthened element switched from a fracture inside the substrate to adhesive failure as the water exposure time increased (this failure mode was also observed for specimens subjected to wet-dry cycles at 25 °C [67]). When the specimen was dried after being immersed in water, it showed a 13% flexural strength recovery. Laminate specimens showed the worst performance of all. However, the lack of test results and the several possible strengthening schemes make it difficult to define time or degradation ranges for this condition.

### 3.2. Saline Exposure

Civil structures exposed to marine environments or deicing salts during the winter are generally subjected to saline environments. Studies have found that the intermolecular bonds in the matrix as well as in the fiber-matrix interface could strain or rupture in the presence of salts [68]. Similar to the effects of water exposure, mechanical properties deterioration of polymer resins when exposed to saline solutions is influenced by the ingress of humidity, which may lead to irreversible changes in the polymer matrix (i.e., plasticization and hydrolysis) [69].

GFRP bars

GFRP bars immersed in a saline solution do not necessarily have a significant difference in strength and stiffness when compared to those in a solution without salt [70,71]. Laboratory tests (see Figure 3) with different solution concentrations have shown a tensile strength retention of approximately 88% for GFRP bars at 60 °C [14], with no significant degradation of the modulus of elasticity. High-volume fiber content has been found to affect the absorption of seawater [72]. Furthermore, severe degradation could be found due to the coupling action of the salt and alkali environment [55]. Studies of GFRP bars embedded in concrete have resulted in long-term prediction (10 years) degradation models (based on the Arrhenius theory) of the tensile strength capacity of 92% and 72% for typical field exposure and aggressive exposure (i.e., seawater at 60 °C), respectively [73]. Other models for service life up to the 100 years predicted tensile-strength retention of about 70% [74]. Pull-out tests on GFRP bars embedded in conventional and seawater-mixed concrete at 24 months showed some differences in performance, with a maximum reduction of 11% for the seawater specimen [75]. Khatibmasjedi [75] stated that it is impossible to make generic statements about all bars given the data scatter observed in the literature. D’Antino & Pisani [60] analyzed 47 tensile tests of bars immersed in salt solutions and found an average residual strength ratio of 88.2%, 87.0% and 68.2% for temperatures between 11–25 °C, 40–50 °C and 80 °C), respectively. Al-Salloum et al. [56], Duo et al. [14], and D’Antino & Pisani [60] can be consulted for a more comprehensive analysis of the impact of saline exposure on GFRP bars.

CFRP fabrics

FRP strengthening systems may go through a significant reduction in bond strength between the FRP and the bonded surface when exposed to saline environments [46]. The durability of the FRP system can be strongly influenced by the resin properties, most of which absorb between 1% and 7% moisture by weight. Li et al. [76] found that, due to the higher NaCl concentration in wet dry-cycles compared to immersion exposure, the former has a stronger effect on the performance of the strengthened elements. Al Nuaimi et al. [77] tested RC-beams specimens strengthened with externally bonded CFRP fabrics (epoxy matrix), observing a change in the failure modes from cohesive to adhesive in saline water exposure, indicating the bond strength critical aspect in hygrothermal exposure. Furthermore, the tested specimens recovered some strength after 180 days of exposure, revealing that, with age, the epoxy matrix matures by forming more cross-links, making possible a slightly higher mechanical performance [77]. The degree of retention of flexural strength after ageing relies on the type of laminate whose deterioration is primarily due to physical processes (e.g., plasticization of the polymeric matrix) and not a chemical degradation [78]. Li et al. [76] and Al Nuaimi et al. [77] can be consulted for a more detailed analysis.

### 3.3. Alkaline Exposure

Given the alkalinity environment of concrete, FRP composites could be affected when embedded in it or when exposed to an alkaline environment. Among the available fiber types, carbon fibers are resistant to alkalis and do not degrade as much as glass and basalt fibers [79]. Silica, the primary component of both basalt and glass fibers, is vulnerable to chemical attacks, and hence defines the chemical degradation mechanisms of both BFRPs and GFRPs [79,80].

GFRP bars

Alkaline solutions with pH values ranging between 11.5 and 13.0 degrade the tensile strength and stiffness of GFRP bars [81]. Benmokrane et al. [82] immersed GFRP bars in an alkaline solution for up to 5000 h at 60 °C to study the durability of the bars made with a vinyl-ester resin by assessing the transverse-shear strength (a critical parameter for dowels in pavements and longitudinal bars in reinforced concrete beams subject to shear cracking), the flexural strength and the interlaminar shear strength (governed by the fiber/matrix interface). All properties were found to have a strength retention of at least more than 70%. Figure 4 shows the results in terms of the tensile strength retention of GFRP bars when subjected to an alkali solution from the available literature [14,55,57,58,83,84,85]. The average strength retention ratio at 20 °C was found to be 89%, while the modulus of elasticity did not show significant deterioration. Compared to elevated temperatures, at room temperature (20 °C) the rate of degradation is slower with increasing exposure time. Initial studies stated that micro-cracks generated on the surface of the polymer matrix when the GFRP rebars are stressed allow the alkaline solution to enter, ultimately leading to fracture of the rebar [52]. However, recent studies have observed that sustained stress has limited the influence in the tensile strength retention of GFRP rebars when exposed to alkaline environments [86]. D’Antino & Pisani [60] analyzed 202 tensile tests of GFRP bars immersed in alkaline solutions and found an average residual strength ratio of 88.9%, 84.3%, and 73.2% for temperatures between 11–25 °C, 26–53 °C, and 57–80 °C, respectively. Duo et al. [14], Al-Salloum et al. [56] and D’Antino & Pisani [60] provide a more detailed analysis of the exposure to alkaline environments.

Robert et al. [84] compared GFRP bars embedded in mortar and immersed in tap water with rebars subjected to a simulated pore-water solution, finding less degradation by accelerated aging in the former and highlighting the conservative approach when designing with GFRP rebars. Nepomuceno et al. [61] reviewed 72 tests of GFRP bars immersed in alkaline solution, observing a bond strength retention of 81% at the lower bound.

Furthermore, due to the diffusion of carbon dioxide from the atmosphere and the ingress of chloride ions, the natural alkalinity within the concrete is lost. In RC-elements reinforced with steel, this can result in a reduction of the pH, which breaks the passive layer of iron oxide around the steel rebar, and consequently the steel can start corroding [87,88]. However, carbonation has been proven to increase the strength of carbonated concrete, having a favorable impact on concrete strength [89]. Demis and Papadakis [90] found no substantial bond deterioration until the carbonation reaches the FRP bar. Overall, the increase in concrete strength counteracts the bond deterioration due to carbonation in the FRP RC elements.

CFRP fabrics

As it relates to external FRP applications, some laboratory tests have indicated that the tensile properties of CFRP fabrics are barely affected by alkaline solutions, although a reduction in flexural strength was observed [91]. The flexural strength retention has been found to be thickness dependent: the thicker the CFRP laminate the higher the retention [92]. The reduction in load carrying capacity of CFRP specimens subjected to alkaline solution at room temperature were found to be dependent on the strengthening scheme used, varying from 4.7–23.3%. With an increase of temperature [46] to 60 °C, the change in bond strength showed a slower rate of reduction [93]. CFRP samples taken from a 12-year-old existing retrofitted bridge showed degradation in the modulus of elasticity and the tensile strength, which were 3% and 21% lower than those of new samples [93].

### 3.4. UV Exposure

The ultraviolet (UV) exposure may result in surface oxidation due to different chemical mechanisms related to the resin type [94,95]. UV radiation can degrade the molecular bonds in the polymer matrix and hence damage the FRP composite [96]. Although the UV exposure can degrade only the top few microns of the surface, the damaged area could be the point of stress concentration and might invite other environmental attacks.

GFRP bars

Tests on GFRP bars showed a tensile strength loss after 3 years of exposure of about 10% [97]. Additional studies have shown an 8% reduction on GFRP rods after 500 h (no reduction thereafter) [98]. Additional studies have shown no significant degradation of GFRP bars when exposed to UV lights [99]. The FRP rebars embedded in concrete are themselves protected from UV exposure, but they are vulnerable during storage or when used as external reinforcements [100,101]. A UV-resistant coating or paint can be applied to the surface of the composite or the addition of UV stabilizers to the composite resin during the manufacturing process can also help to reduce the effects of UV radiation on the material. Ultraviolet stabilizers could be added to protect the resin from the effects of sunlight [102].

CFRP fabrics

On the other hand, the effect of ultraviolet (UV) radiation is more relevant in external applications, as the CFRP fabric will be directly exposed to UV lights [103,104]. In FRP composites, fibers (i.e., carbon fibers) are less vulnerable to UV radiation; in contrast, most resins will be impacted by UV radiation [105]. This could be avoided by structural measures or material modifications, such as additional matrix additives, pigmented gel coatings, or painting after installation. According to the CSA S806 [20] “UV exposure can cause embrittlement and micro-cracking in an unprotected laminate surface”. Color shift or yellowing and gloss changes are some of the effects of UV exposure on FRP laminates. The ultimate tensile strength and modulus of elasticity of the FRP composite may be affected when subjected to UV radiation because of the degradation of the resin adhesive (the main degradation parameter [106]. This is because increasing the number of FRP layers may not help to reduce the UV exposure degradation. Beam specimens strengthened with CFRP fabrics showed a slightly higher stiffness after 360 and 730 days compared with those at 180 days after UV exposure [77]. Zhao et al. [106] includes a more detailed analysis of exposure to ultraviolet exposure.

### 3.5. Freeze-Thaw Exposure

Freeze-thaw cycles along with the coupled effect of high moisture may lead to the deterioration of FRP composites [9,107]. FRP materials become brittle under freezing temperatures, and the moisture absorbed expands when it freezes, leading to microcracks, particularly at the interface between the FRP and the substrate material [100,108]. Moreover, ice crystal growth during repeated freeze-thaw cycles generates micro-cracking due to the high pressure in the concrete, reducing the bond strength between them [3,109]. These micro/macro-cracks could accelerate degradation by allowing the penetration of other chemical solutions. However, the extent of damage in concrete resulting from freeze-thaw cycles is influenced by its saturation level. Dry concrete is relatively resistant to freeze-thaw cycles and is minimally impacted, even with a relative humidity of 75–80% [3].

GFRP bars

Experimental studies on FRP bars have shown different performances when exposed to freeze-thaw cycles. Pull-out test results of GFRP bars (freeze-thaw cycles ranging from 50 to 600) showed that the bond strength between the FRP bar and concrete surface was not considerably influenced by the environmental conditions [110]. Other tests in GFRP bars subjected to as many as 300 freeze-thaw cycles have shown a 20% decrease in the bond resistance in pull-out tests, and no more than a 10% deterioration in tensile strength [95,97,111,112].

The results on sand coated GFRP reinforced concrete (6% entrained air content) beams exposed up to 360 freeze/thaw cycles (−20 to 20 °C), either in an unstressed state or loaded in bending, showed no significant effect on the behavior in terms of deflections, strains, or ultimate capacity for the two actions when compared to control specimens [113]. Some specimens showed an even better resistance after the test, which might be explained by the increase in concrete strength during conditioning (50% humidity). Alves et al. [114] indicated that the effect of fatigue loading was more pronounced than that of freeze-thaw cycles when the bond performance of the sand-coated GFRP/vinyl ester bars embedded in concrete subjected to freeze-thaw cycles combined with sustained axial load and fatigue loading was examined. In fact, freeze-thaw cycles combined with a sustained load increased the bond strength by around 40%. This increase was attributed to the GFRP bar absorbing moisture and expanding in the cross-sectional area, thus enhancing the friction mechanism.

CFRP fabrics

Progressive degradation of the FRP composite and the weakening of the interfacial bond at the FRP to concrete interface can occur as a result of freeze-thaw cycles [46]. Chajes et al. [115] reported a tensile strength loss for CFRP-strengthened beams of 9% when exposed to freeze-thaw cycles compared to either GFRP or AFRP, which showed losses of about 50%. Karbhari and Zhao [116] also reported the better performance of CFRP over the GFRP strengthening system, stating that freeze-thaw cycles can cause matrix hardening, microcracking, and overall bond degradation. Other studies have reported no changes in the strength performance when exposed to freeze-thaw cycles [46,117].

Homam and Sheikh [118] immersed CFRP coupons in water and then subjected them in up to 300 freeze-thaw cycles between –18 °C and 4 °C. The tensile strength was not significantly affected by the exposure. RC beams strengthened with externally bonded CFRP plates in flexure subjected to up to 200 freeze–thaw cycles [119] showed no significant adverse effects.

### 3.6. Elevated Temperature and Fire Exposure

Due to the anisotropic behavior of FRP composites, the coefficient of thermal expansion (CTE) is different in longitudinal and transversal directions; the first one controlled by the fibers and the second one controlled by the resin [120,121]. In addition, the CTE also differs between the concrete and the FRP composite, being 5–8 times higher along the transverse direction [100]. At temperatures above 300–400 °C (572–752 °F), the thermal decomposition of the FRP organic matrix usually occurs with the potential emission of smoke, soot, toxic/combustible volatiles, and heat. Even though concrete usually has a high resistance to fire, its mechanical properties could also degrade when exposed to elevated temperatures by accelerating the environmental effects, as discussed above.

GFRP bars

Tests on GFRP bars have evidenced a reduction in their tensile strength and bond properties at temperatures between 100 °C and 350 °C [122,123]. Tensile tests carried out on ECR-glass FRP (vinyl ester hybrid resin) bars exposed to elevated temperatures indicate a limit for temperature ranging from 400 to 450 °C, corresponding to a tensile strength loss of around 30% to 55% [124]. The bond strength between the FRP bar and concrete showed a substantial decrease near 180 °C (close to the glass transition temperature, Tg). Hajiloo & Green [125] carried out pull-out tests on 16 mm GFRP bars (sand-coated, sand-coated braided, and ribbed with an average of 84% fiber volume and Tg ≈ 120 °C) under steady-state and transient temperature protocols (temperatures ranging from 25 to 360 °C). The tests showed that when the concrete-to-bar interface temperature reached 75 °C, the bond strength losses were approximately 27%, stating also that the bond strength between the FRP bar and concrete at high temperatures is strongly related to the glass transition temperature of the resin. Similarly, under high temperature above the glass transition temperature, the resin matrix softens and transitions into a rubber like material which reduces the matrix stiffness and degrades the fiber-matrix interface [9]. These losses in bond strength could lead to splitting cracks affecting the element load capacity. Aiello et al. [126] stated that a cover value greater than two times the bar diameter could avoid the occurrence of the splitting cracks when subjected to a temperature increase of 50 °C.

Related to fire exposure, a study carried out by Kodur and Bisby [127] showed that GFRP-reinforced slabs (no information was provided on the fibers and resin) have lower fire resistance than steel reinforced slabs in terms of the critical temperature for the reinforcement (described as those at which the FRP bar lost 50% of its ultimate tensile strength), and are influenced by the aggregate type and concrete cover. After four hours of exposure to fire, spalling was not observed in any of the slabs. They concluded that the heat transfer behavior of FRP–RC slabs appears similar to steel-RC slabs, but the FRP-RC slabs have much lower fire resistance. However, this was based on assumed critical temperatures of 250 °C for GFRP specimens. The study did not account for thermally induced bond degradation [128].

Nigro et al. [129] tested four GFRP reinforced concrete slabs under typical design loads exposed to fire action. The results indicated that temperatures of up to 460 °C may be reached with a failure associated to the rupture of fibers in the middle of the element if anchorage length (i.e., 500 mm) outside the exposed zone is provided to avoid the pull-out of the bar. This anchorage length ensures the resistance beyond the glass transition temperature (Tg) in the exposure zone when the resin softening reduces the adhesion between the FRP and concrete. A bent end may reduce the anchorage length. However, the bar could still become unbonded and larger cracks could appear. If a sufficient anchorage length is not provided, the fire endurance will depend on the slippage of the bar at temperatures of around Tg [130]. Table 2 shows the relevant results on studies related to FRP-RC elements exposed to elevated temperatures, as discussed previously. The impact on the mechanical properties in FRP bars embedded in concrete exposed to fire or elevated temperature is quite evident. Extended bar anchorage and deeper concrete cover could improve the fire performance of the concrete element. However, the overall performance depends on several parameters related to the bar constituents.

CFRP fabrics

Moderate ranges of temperature variation of around ±28 °C (±50 °F) has been shown not to affect the bond between the surfaces of FRP strengthening elements [46]. A study carried out in Switzerland [136] on the fire performance of RC-beams strengthened in flexure with external CFRP strips, observed a fire endurance of 81 min for unprotected specimens and 146 min for specimens with a thermal insulation. Bisby et al. [137] demonstrated that sufficient fire endurance can be provided to CFRP-strengthened reinforced concrete slabs with adequate insulation. The slab tests suggested that even with 38 mm of supplemental insulation, it will be very difficult to prevent FRP temperature from exceeding the glass transition temperature of the resin for more than one hour. Other studies on T-beams strengthened with externally bonded CFRP fabrics [138] determined that one layer of VG insulation (vermiculite gypsum—a lightweight, fire-resistant cementitious plaster that can be spray-applied) of 25 and 38 mm can achieve a fire endurance of more than 4 h (based on ASTM E119); the Tg of the epoxy was exceeded between 16 and 36 min and between 55 and 57 min for the 25 and 38 mm of insulation, respectively. The latter kept the average FRP temperature below the matrix Tg for 54 min.

## 4. Mechanical Effects

Mechanical effects such as cyclic fatigue, creep (static fatigue), or relaxation could affect the properties of the FRP composite and impact the serviceability performance of the FRP-RSC element. This mainly occurs in internal applications due to FRP characteristics that may control the design and cause structural elements to perform differently under time-dependent phenomena.

### 4.1. Cyclic Fatigue

Cyclic fatigue, commonly called fatigue, is a degradation of the capacity of a material caused by repeated applications of a large number of load cycles [98]. Most of the available data related to fatigue behavior of stand-alone FRP materials was obtained from aerospace applications [13]. However, tests have been carried out on FRP bars and laminates to study the impact of repeated load cycles in FRP composites. Its behavior is different from that of metal that occurs by the initiation of single crack. In FRP composites, an accumulation of damage controls the fatigue resistance that includes fiber-matrix debonding, matrix cracking, delamination, and fiber fracture [139]. Under tension-tension fatigue, fiber rupture is the primary failure mode in unidirectional composites with linear-behavior up to failure [98].

GFRP bars

Initial studies in GFRP bars observed a cyclic tensile fatigue effect of an approximately 10% loss in the initial static capacity per decade of logarithmic lifetime [13]. However, when embedded in concrete, GFRP bars may exhibit a shorter fatigue life than those of bare GFRP in air [140]. This is mainly attributed to the abrasion between the GFRP-to-concrete surfaces. This effect has been studied without any conclusive results, and this is explained by the differences in materials, environments, and test methods [13]. Alves et al. [114] found a reduction of at least 30% in GFRP bars when subjected to one million cycles at 25% of the guaranteed tensile strength (GTS) in pull-out tests. Other studies on slab elements have found the rate of degradation to be comparable to elements reinforced with steel [141,142]. Table 3 shows the recent fatigue studies on GFRP-RC elements together with their main findings.

CFRP fabrics

The various modes of fatigue failure in CFRP laminates have been recognized as comprising matrix cracking, transverse cracking, interfacial debonding, delamination, and fiber breaking. The occurrence of failure is influenced by factors such as the types of fibers and resins used, the structure of the textile, the surface treatments applied, the fatigue stress ratios, and stress levels [145,146]. Initial studies on RC-beams strengthened with externally bonded CFRP laminates under fatigue loading showed that the performance of the strengthened beam subjected service-load conditions that continued to be controlled by the original steel reinforcement design (with increase in deflection and loss of initial stiffness). However, for overload conditions, damage propagation at the concrete-composite bond interface was observed as the load cycles increased [147]. Ferrier et al. [148] observed a redistribution of stresses from the FRP laminate to the steel rebars during fatigue. In most cases, the failure has been observed to be initiated by the successive yielding of the steel rebars in tension, leading to debonding of the CFRP laminate, the latter considered to be a secondary failure mode [149]. The same authors found that one layer of FRP laminate would increase the fatigue service load by 40% compared to a non-strengthened RC beam (to achieve this, rebar yielding in RC beams should not be permitted).

Zheng et al. [150] examined the fatigue performance of bonds between the carbon fiber laminate (CFL, pre-saturated laminates with epoxy adhesive) and concrete by carrying out seventeen double shear tests at 60 °C and 95% RH. The results showed a negative influence of the exposure on the bond performance and the fatigue life. The authors stated that fatigue life (in terms of cycles of loading) may also be reduced by a higher stress level.

### 4.2. Creep Rupture

Static fatigue or creep rupture is a phenomenon produced by constant tension over time that can lead to sudden failure after a period called the “endurance time” [13]. In FRP composites, a thermoplastic matrix is commonly not used in structural engineering applications due to its low creep resistance. On the other hand, a thermosetting matrix (polyesters, epoxies, and vinyl esters) has a higher creep resistance, and is the matrix used in structural applications [8,151]. Endurance time may be shortened by combined effects, such as harsh environmental conditions and a high sustained stress-to-strength ratio [12,152,153]. Moisture and fluid absorption decrease the creep rate for concrete but increase the creep rate in FRP bars, causing residual stresses and degrading polymers, fibers and fiber/matrix interfaces via hydrolysis and chemical attacks. Singhvi & Mirmiran [71] stated that the presence of salt in the water solution does not affect the creep rate of FRP-RC in the same way as the moisture does. The susceptibility of the fiber to creep depends on the material, with glass fibers being the most susceptible [98].

GFRP bars

Initial studies on GFRP bars observed a creep strain under a sustained load at 80% of the ultimate tensile strength (UTS) of the GFRP bar after 50 days, ranging from 0.3 to 1.0%. Other studies observed a creep strength rupture equal to 55% of the UTS for an extrapolated endurance time of 50 years [98]. It was observed that a linear relationship existed between the creep rupture strength and the logarithm of time for a period of up to 100 h. The results were extrapolated to 57 years, yielding a linear extrapolation of the ratio of creep rupture strength to the UTS of bars to be 0.29 [98]. Additionally, Laoubi et al. [113] found the creep strain on GFRP bars to be less than 2.0% of the initial value after 180 days under 27% of the ultimate tensile strength (UTS). More recently, Benmokrane et al. [154] collected and evaluated 204 creep-rupture tests and proposed a value of 50.7% of the average UTS for the creep-rupture strength at a 114-year endurance time for GFRP bars of sizes varying from 6 to 16 mm in diameter. The results of time-dependent bond slip tests on GFRP pull-out specimens due to sustained loading (15% of UTS) showed an increase in the bond slip that stabilizes at approximately 60 days after loading, irrespective of the concrete strength. However, the concrete compressive strength influences the bond deterioration, and the greater the concrete strength the lower the creep slip [155]. Table 4 shows the recent studies on creep in GFRP-RC elements.

CFRP fabrics

The shear stress to shear strength ratio is a primary factor affecting the long-term behavior of the concrete-CFRP interfaces. Results focused on the epoxy resin have indicated an ultimate creep coefficient (the ratio between the time dependent creep strain and the instantaneous elastic strain) of 3.0, and a creep retardation time (the time when 63% of the creep has occurred) between 1 and 2 days compared to the typical retardation time of concrete that ranges between 300 and 900 days [156]. Results have shown that creep behavior is correlated with the glass transition temperature of the polymer [157]. High shear moduli (>10 GPa) and high glass transition temperature (>55 °C) polymer adhesives are recommended to limit creep in the adhesive joint.

Externally bonded systems are generally used for strengthening applications when an element is expected to carry increased service loads. Al Chami et al. [158] tested CFRP-strengthened concrete beams (laminates with epoxy adhesive) subjected for about one year to sustained loads varying from 59% to 78% of the ultimate static capacities of the un-strengthened beams (in order to account for the increased service loads), finding that the CFRP strengthening system increased the ultimate capacity of the beams, but that there was no significant improvement in creep behavior.

### 4.3. Shrinkage

When manufacturing FRP composites, the resin shrinks during the curing cycle. This occurs due to a chemical loss of volume followed by thermal contraction during cool-down after cure [159]. Shrinkage volumetric reduction is in the range of 5–12%, 5–10%, and less than 5% for polyester, vinyl ester and epoxy resins, respectively. This phenomenon could lead to porosity or micro-cracks [160]. The presence of voids and cracks are considered detrimental to the structural integrity of the composite and may cause reduced electrical resistance [161]. ASTM D5117 establishes a test method to observe the wicking action through delamination or longitudinal continuous voids. However, no specific values are indicated to imply an unsatisfactory performance of the composite.

Shao and Mirmiran [162] tested five different configurations of CFRP grids for the control of plastic shrinkage cracking of concrete using CFRP rods. Test results showed that carbon FRP grids meet or exceed the acceptance criteria (according to ICC-ES [163] for synthetic fibers) for providing a minimum 40% reduction in shrinkage cracking for concrete. However, there is still a lack of information and guidelines related to shrinkage and secondary reinforcement configurations that uses FRP material.

## 5. Impact on Design Requirements of GFRP-RSC Structures

### 5.1. FRP Reinforced Concrete Structures

Deflections in FRP-RC elements tend to be greater in magnitude with comparable steel reinforcement because of the lower stiffness associated with commercially available FRP reinforcement. FRP reinforced flexural concrete elements typically go through extensive cracking and large deflections prior to failure, which could be considered a warning for failure similar to the concept of ductility for steel-RC elements [12]. However, the new generation of FRP rebars has brought improvements in the properties of the rebars [56,164], such as the increase in the value of the modulus of elasticity (e.g., GFRP rebars with modulus of elasticity ranging from 44.8 GPa (6500 ksi) to 60 GPa (8700 ksi) are commercially available), which will have a positive impact in the overall performance of the element.

FRP-RC members are more sensitive to the parameters affecting deflection. Generally, the FRP design for flexural strength may not satisfy serviceability criteria for deflection and crack control [165,166,167], and consequently, serviceability limit states (i.e., deflection, crack control, creep, fatigue) can control the design of FRP-RC members. FRP-RC flexural elements have a relatively smaller stiffness after cracking when compared to steel-RC members of identical size and reinforcement layout [13]. Furthermore, an equal area of FRP will result in larger deflections and wider cracks in the FRP-RC element. The neutral axis depth for the balanced section in the FRP-RC member is close to the compressive end, and higher compressive strains in the concrete are expected in the FRP reinforced section for the same beam depth and applied moment as the steel-reinforced section [168].

At present, most of the available design guides and recommendations for FRP reinforced concrete elements are focused on GFRP rebars [15,17,18,20]. Given the brittle behavior of both FRP reinforcement and concrete; compression- and tension-controlled sections are acceptable in the design of flexural members reinforced with FRP bars, with a strength reduction factor ranging from 0.65 to 0.55, respectively [15]. This differs from steel-RC design, where the compression-controlled sections have a lower strength reduction factor (0.65–0.75) compared to tension-controlled sections (0.90). Table 5 shows the strength reduction factors for different design guides and codes for FRP- and steel-reinforced concrete elements. The serviceability criteria for crack width and deflections are generally satisfied when a section is designed to achieve concrete crushing failure prior to tensile rupture of the FRP rebar [12].

#### 5.1.1. Reduction Factor for Environmental Conditions

The strength and stiffness of the FRP composite may change under harsh environmental conditions and could limit the service life of the structure. Even though FRP rebars do not exhibit corrosion the same way as steel rebars, as referenced earlier, long-term exposure to harsh environments such as moisture and alkaline solutions may deteriorate them and lead to a loss of tensile and bond strength. Furthermore, unlike the visible warning signs of steel-RC elements (e.g., cracking or spalling of the concrete), the deterioration of FRP rebar does not result in a noticeable increase in volume and hence does not lead to the cracking and/or spalling of the concrete cover [56,171]. It is therefore evident that the durability requirements for FRP-RC structures differ from those utilizing steel-RC because of the unique properties of FRP bars. Consequently, some of the design criteria that aims at mitigating the corrosion of the conventional rebar such as increased concrete cover, the use of corrosion inhibiting admixtures, the use of epoxy coatings, and limitations on crack widths to delay the initiation of corrosion are not applicable to or necessary for FRP-RC structures [15]. Unlike steel–RC structural members, for FRP–RC members, there is no special concrete cover thickness provision required for corrosion control [127].

The absence of warning for failure of the FRP-RC element when exposed to harsh environmental conditions can be seen as a deficiency for the FRP-RC members. This is mainly because of the brittle failure characteristic of FRP [172]. To address these deficiencies, design guides for FRP-RC establish the design tensile strength of FRP bars [*f_fu_*] as the guaranteed tensile strength [*f_fu_**] (value reported by the manufacturer computed as no larger than the mean [*f_u_*] minus three standards deviations) times an environmental reduction factor [*C_E_*] to account for the long-term environmental degradation of GFRP bars in service, as shown in Equation (1) and Table 6 for different design guides. The previous ACI 440.1R-15 [13] established a value between 0.7–0.8 as AASHTO GFRP specifications [18]. The new 0.85 value was determined based on 361 accelerated aging tests of unstressed bars [173].
(1)$$f_{fu}=C_{E}f_{fu}{}^{*}$$

AASHTO [18] states that durability conditions related to concrete remain the same as those addressed in steel-reinforced concrete elements, and they can be found in the AASHTO LRFD Bridge Design Specifications [170]. The new ACI GFRP code [15] also establishes durability requirements for concrete mixes in Chapters 19.3.2 and 26.4 to account for applicable environmental exposure.

#### 5.1.2. Fire Protection

The critical glass transition temperature of FRP is generally much lower than that which causes a loss of stiffness in steel. The impact of elevated temperatures of FRP composites is swift and leads to a severe decline in both cross-sectional and bond properties. The critical temperature for tensile reinforcement is traditionally defined as “the temperature at which it experiences a significant loss of strength (50% loss of room temperature yield strength) and becomes unable to sustain applied load”. For reference, this temperature is 593 °C for reinforcing steel [174]. However, for FRP composites there is no specific critical temperature value. AASHTO [18] specifies a minimum concrete cover for GFRP bars ranging from 1.0 to 2.0 times the bar diameters for all exposure conditions except for additional fire protection. However, no additional recommendation on the latter has been established. As mentioned earlier for the effect of the elevated temperature exposure, if GFRP bars are anchored well outside of the area directly exposed to fire, they can retain considerable strength and stiffness during a fire event [130]. ACI 440.11 code [15] also establishes a minimum concrete cover for cast-in-place and precast concrete members in its table 20.5.1.3.1 with a complementary table in the comments section with a fire resistance rating provided by the minimum covers.

#### 5.1.3. Creep and Fatigue Limits

To prevent the failure of GFRP reinforcing bars due to creep, the sustained stress must be restricted to the creep rupture stress. This will ensure that the stress levels remain within the elastic range of the member (an elastic analysis can be used for computation of the stresses) [18]. Determining the applied stress by considering the full live load is overly conservative. To account for only the sustained portion of the live load, the load factor applied to it is reduced from 1.0 to 0.2 [12]. It is important to note that for conventional steel-RC, no additional long-term considerations are necessary, as steel does not experience either shrinkage or creep [155]. Furthermore, the maximum sustained tensile stress in the FRP reinforcement shall be less than the design tensile strength of FRP reinforcing bars times a creep rupture reduction factor (equal to 0.30 following ASTM D7337/D7337M, in the previous ACI 440.1R-15 guide [13], this value was equal to 0.2 for GFRP bars) [18]. Table 7 shows the creep rupture reduction factor for different design guides and codes for FRP and steel design. Although the use of GFRP bars as a compression reinforcement of flexural members is not recommended, placing GFRP rebars in the compression zone of flexural members is permitted, providing that they are not considered for the determination of the member flexural resistance [18]. This is because the limited compressive strength and modulus of the FRP bars do not increase the strength nor reduce the effect of concrete creep of FRP reinforced flexural members [12].

#### 5.1.4. Crack Width and Deflection

Concrete elements subjected to any load condition resulting in tensile stresses are prone to cracking and, in order to provide a good quality appearance, crack control provisions are aimed at controlling the crack width through the rebar distribution. Moreover, the crack width limits for FRP reinforced concrete elements under service are recommended to limit fluid ingress that could degrade the FRP bars while also ensuring their acceptable aesthetic appearance [102]. The maximum crack width of FRP reinforced concrete is limited to 0.7112 mm (0.028 in.) [15] to address concerns related to durability (this limit is set at 0.4572 mm (0.018 in.) for steel-RC structures). In addition to this requirement, for GFRP-RC elements a stress at service limit is set to similarly control the cracks.

Shrinkage in concrete can also influence the crack width and, therefore, minimum shrinkage and temperature reinforcement provisions are specified in the design guidelines based on the 0.0018 reinforcement ratio required by ACI 318 [169] for Grade 60 steel. This is due to the fact that there is insufficient data available for the minimum GFRP reinforcement ratio for shrinkage and temperature; and although empirical, the steel approach has been used satisfactorily for many years [15,18]. It is worth mentioning that this chapter does not include all the serviceability requirements stipulated in the different design codes.

### 5.2. FRP Strengthened RC-Concrete Structures

FRP laminates along the tension face of RC elements is an effective method to increase the flexural strength where different schemes may be used to install the laminate/sheets [44]. Flexural strengthening has proved to be an effective method to limit the crack propagation and the crack width in the flexural zone by increasing its ultimate strength and decreasing the initial deflection [158]. Generally, the deformation under service load of flexural members with or without FRP strengthening is small, and deflections in this range can be estimated using a transformed-section analysis where the tensile contribution of the FRP is added to the contribution of the steel reinforcement [7,102]. Typically, the contribution of the FRP takes place after the longitudinal reinforcing steel has yielded.

The serviceability limits of the externally strengthened concrete elements are usually limited by the original design (typically steel-RC) and the condition of the RC element substrate [147]. The available AASHTO design specifications [19] (published 10 years ago) for externally bonded FRP systems allow the use of wet layup and precured systems. ACI 440.2R [16], on the other hand, also allows the use of prepeg and NSM systems. Similar to FRP-RC structures, the strength and strain of the FRP composite in ACI 440.2R is affected by an environmental reduction factor to account for long-term exposure, as shown in Table 8. No environmental reduction factor is found in the AASHTO specifications [19].

In order to avoid inelastic deformations in the existing steel-RC element, the stress in the steel rebars under service load is limited to 80% of the yield strength [16,19]. In addition, the concrete compressive stress is also limited to 60% and 40% in ACI and AASHTO, respectively. AASHTO accounts for creep rupture by limiting the FRP strain to the FRP tensile failure strain times a strain limitation coefficient of 0.55, 0.3 and 0.2 for CFRP, AFRP and GFRP, respectively. On the other hand, ACI limits the FRP stresses (to account for creep rupture and fatigue stress) to 0.55, 0.3 and 0.2 of the design ultimate tensile strength of FRP for CFRP, AFRP and GFRP, respectively.

## 6. Summary and Conclusions

### 6.1. Summary

As previously discussed, harsh environmental conditions may affect the mechanical properties of the FRP composite (degrading the fiber, the matrix, and the interface between them) [53]. Table 9 shows the performance summary of FRP composites under harsh environmental conditions based on Benmokrane’s [11] criteria, which is qualitative and only depends on the tensile strength degradation, being “excellent” if it is less than 10%, “good” if it is between 10% and 20%, “moderate” if it is between 20% and 30%, and “poor” if is larger than 30%.

GFRP bars

Overall, moisture and alkaline environmental conditions have a more degrading influence on GFRP bars than the other environmental factors [55]. When GFRP bars are immersed in water, the polymer matrix absorbs the water (or alkaline in the water) and may cause delamination and cracks at the interface between the fiber and the matrix, which speeds up the diffusion of water and chemicals [52]. The deterioration of glass fibers is caused by etching and leaching, which can lead to hydrolysis (breaking and weakening of the polymer chains), plasticization (softening of the resin and thus reduced stiffness), and swelling, all resulting in the degradation of the polymeric matrix [55]. The degradation rate depends on the rate of fluid absorption [95]. To consider their long-term influence in the GFRP rebar, the design of GFRP-RC elements (based on ACI 440.11) includes an environmental factor of 0.85 (which was established assuming a service life of 75–100 years).

In addition, the mechanical effects also have a significant impact, since the serviceability limit state typically controls the flexural design. Although a similar approach to that of steel-RC structures is used, there are different requirements, such as the creep and fatigue limits, and its limitation for use as a compressive reinforcement. The ACI code establishes a sustained load of a maximum 30% of the design tensile strength (which already includes the environmental factor of 0.85). This value includes a 1.67 safety factor and was derived based on creep-rupture tests that projected a 50.7% of the average UTS for a 114-year endurance time [154]. The creep strain in the GFRP bars was found to be less than 2.0% of the initial value in the early stages. Furthermore, the fatigue resistance of stand-alone FRP composites is controlled by an accumulation of damage that includes fiber-matrix debonding, matrix cracking, delamination, and fiber fracture [139]. However, when embedded in concrete, GFRP bars may exhibit a shorter fatigue life than those of bare GFRP in air [140]. This is mainly attributed to the abrasion between the GFRP-to-concrete surfaces. No conclusive results have been obtained.

CFRP fabrics

On the other hand, for elements strengthened with FRP systems, since they are usually exposed to the weather, the impact can be more severe if adequate protection measures are not taken. The debonding of the interface between the CFRP layer and the concrete is one of the most crucial aspects due to the vulnerability of the resin to harsh environments. In terms of serviceability, the original steel-RC design plays an important role in limiting the stresses and strains in the overall element. When subjected to cyclic loading, it has been observed in most cases (several factors could influence the behavior) that failure is initiated by the successive yielding of the steel rebars in tension, leading to the debonding of the CFRP laminate, the latter of which is considered to be a secondary failure mode [149]. The anchoring performance that the strengthened scheme plays is fundamental to reach the desired capacity and performance of the FRP system. To limit the creep rupture of the FRP composite, the guidelines limit the stresses based on their ultimate tensile strength by approximately 55%.

### 6.2. Conclusions

This paper reviewed the state-of-the-art on the durability and mechanical properties of FRP composites when used for FRP-RSC structures. Glass/Vinyl-ester FRP rebars and Carbon/epoxy FRP fabrics were found to be the most commonly used for internal and external applications, respectively. Additionally, the impact on design requirements was discussed further. It is clear from the data review that as improvements in the material properties are developed and more laboratory results are available, less scattering and more confidence exists to select the appropriate factors to account for long-term exposure. The findings can be summarized as follows:Recent developments in the FRP industry have resulted in improvements in material properties, overcoming some of the past disadvantages. Cost-related limitations are a market- and project size-varying parameter, and it might be priced the same as steel bars. Furthermore, in terms of bendability, the problem could be addressed with the use of thermoplastic resins so that bars can be bent at the fabricator’s shop instead of during pultrusion. Further research is needed.Harsh environmental exposure can affect the mechanical properties of the FRP composites where tensile and bond properties are most commonly affected. The available data focus mainly on GFRP and BFRP bars for internal application and CFRP laminate/sheets for external application. Notwithstanding the information collected by various authors, it remains inadequate for conducting diverse analyses that would incorporate the impact of various parameters. In certain studies, the results of bars have been combined with those of laminates, and the findings of carbon have been mixed with those of glass, which can lead to mixed test outcomes. Furthermore, some disregard the influence of parameters such as fiber content, type of matrix, bar diameter, etc. This is the main reason why a clear degradation pattern cannot be specified, even for a specific composite (e.g., GFRP/vinyl ester bar). Improvements in material properties also affect the prediction models by enhancing the performance under different environmental/mechanical conditions, therefore making the data from early generations of FRP composites unrepresentative of those of today.Alkaline exposure is the most detrimental exposure to the FRP composites. CFRP remains the composite with the best performance in aggressive environments. GFRP shows good performance in different environments. The environmental reduction factors available in the U.S. design guidelines account for the high pH level of both pore-water solutions and the presence of alkali ions, the mean temperature and the humidity, for an assumed service life of 75–100 years. The studies agreed on the need to conduct field research to evaluate and compare the results of short-term laboratory tests.Due to its low modulus of elasticity and its special characteristics, FRP reinforced concrete structures perform differently from steel-RC under time-dependent phenomena. This frequently leads to designs that are controlled by serviceability criteria which can differ from that of steel-RC by having stronger reduction factors applied to the strength and strain of the FRP composite. Some of requirements are based on those of the steel approach. Further studies should be conducted to assess the applicability of these limit requirements. While the design approach between the ACI 440.11-22 (*Code Requirements for Structural Concrete Reinforced with Glass FRP Bars* (1st ed.) [15]) and AASHTO (*LRFD Bridge Design Guide Specifications for GFRP-Reinforced Concrete* (2nd ed.) [18]) for FRP-RC structures it is slightly different with regard to certain requirements, and a review of the requirements for EB-FRP concrete structures reveals several differences in approach between the ACI 440.2R-17 (*Guide for the Design and Construction of Externally Bonded FRP Systems for Strengthening Concrete Structures* [16]) and AASHTO (*Guide Specifications for Design of Bonded FRP Systems for Repair and Strengthening of Concrete Bridge Elements* (1st ed.) [19]). The [19] is becoming outdated, considering that the specifications were first published 10 years ago, and material properties and reinforcement techniques have improved since then.This state of the art review is a starting point for future work aimed at understanding the implications of environmental/mechanical effects on FRP-RSC element properties, as well as the approach used to evaluate serviceability limit states and reinforcement requirements. Furthermore, since the number of FRP structures has been increasing, it is likely that guidelines for the inspection of FRP concrete elements will be required.

## Figures and Tables

**Figure 1 materials-16-01990-f001:**
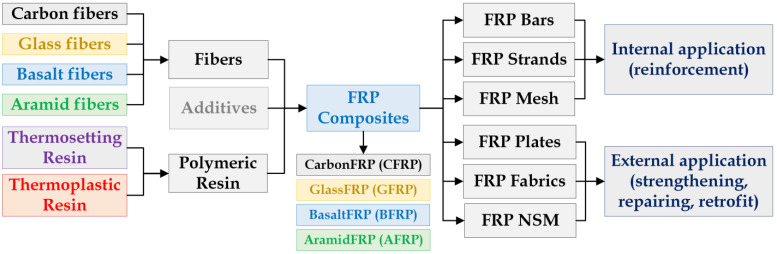
FRP Composite tree diagram for applications in concrete structures.

**Figure 2 materials-16-01990-f002:**
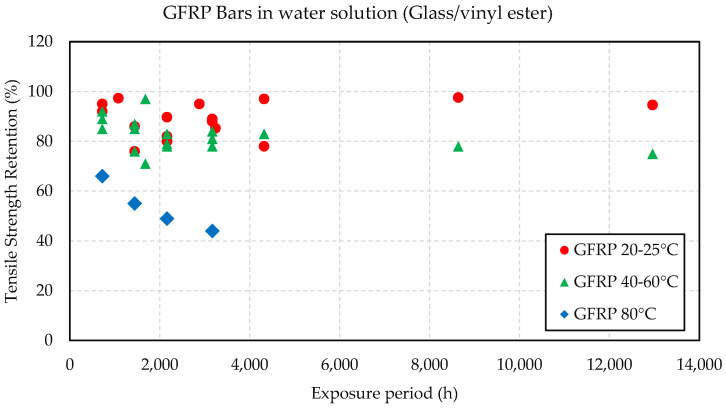
GFRP bars immersed in water solution from the available literature [55,56,57,58,59].

**Figure 3 materials-16-01990-f003:**
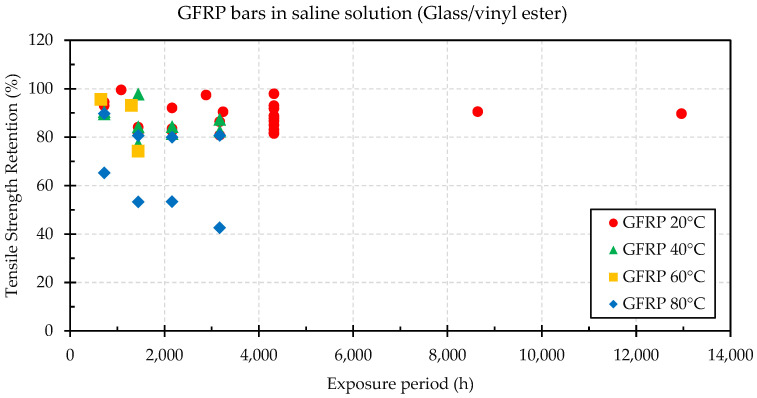
GFRP bars immersed in saline solution (adapted from Duo et al. 2021 [14]).

**Figure 4 materials-16-01990-f004:**
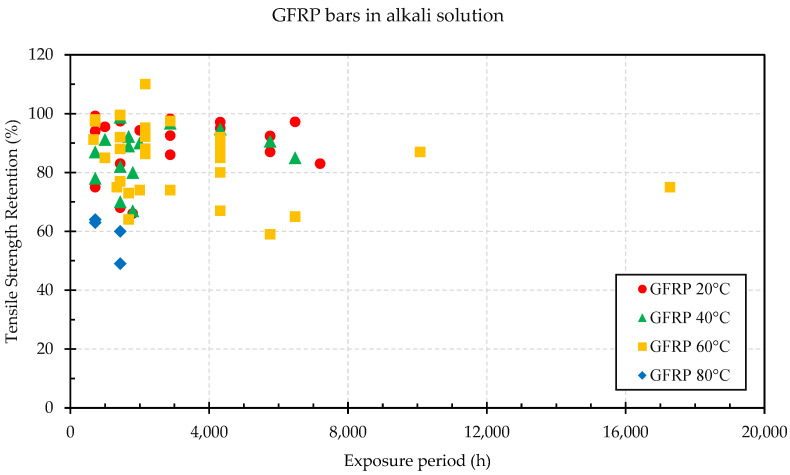
GFRP bars immerse in alkaline solution from available literature [14,55,57,58,83,84,85].

**Table 2 materials-16-01990-t002:** Research studies of FRP-reinforced concrete members exposed to elevated temperatures.

Study	Sample	FRP Type	Tg (°C)	Exposure Condition	Period (min)	Results
[131]	RC Beam (f′c ≈ 42 MPa), 75 mm concrete cover	GFRP (vinyl-ester and polyurethane)	N/A	Fire (460 °C and 380 °C max in rebar)	100–140	Fire tests indicate fire design minimum requirements (90 min) based on BS 476
[128]	200 mm bar (9.5 mm and 12.7 mm diameter)	GFRP (polyester)	N/A	Temperature (100–600 °C)	-	0%–99% tensile strength loss, −17%–72% modulus of elasticity loss
[127]	RC Slab (f′c ≈ 40 MPa), 25.4 mm concrete cover	GFRP/CFRP bars	N/A	Fire (up to 800 °C)	240	50% tensile strength loss, a test fire endurance between 35–45 min
[124]	RC Slab 60 mm concrete cover	GFRP bars	N/A	Fire based on DIN EN 1363	90	bond failure mode when temp in the bars reached approximately 230 °C.
[129]	RC Slab (f′c ≈ 39 MPa), 32–51 mm concrete cover	GFRP bars (70%)	100	Fire based on ISO 834	240	Cover values of 51 mm and anchorage length values of about 500 mm showed better structural behavior
[132]	RC Beam (f′c ≈ 35 MPa), 20 mm concrete cover	GFRP/CFRP bars	N/A	Temperature (up to 900 °C, 500 °C in rebar)	80	CFRP RC beams indicated better stiffness characteristics
[133]	RC Beam (hybrid concrete)	GFRP bars	N/A	Temperature (300–700 °C)	200	GFRP-RC beam capacity was reduced by around 53% at 700 °C.
[134]	RC Beam (f′c ≈ 40 MPa), 30 mm clear cover	BFRP; Hybrid FRP; nano-Hybrid	N/A	Fire, up to 800 °C	80	Reduction in the strength capacity by 43%, 40%, and 43%
[135]	RC Beam (f′c ≈ 41 MPa), 20 mm concrete cover	GFRP/CFRP bars	120	Fire, up to 1000 °C	120	The CFRP beam reached 66% of initial load bearing capacity. The 14 mm GFRP beam reached higher capacities than the 10 mm.

**Table 3 materials-16-01990-t003:** Recent studies on fatigue in GFRP-RC elements.

Study	Materials	Test	Parameters	Condition	Findings
[140]	Sand-coated No.16 bar	Beam-hinge bending test (150 × 200 × 2300 mm)	variable stress range (20 to 55% of GTS)	1,000,000 cycles at 0.2 to 1.0 Hz and 20 to 55% of GTS	Lower fatigue lives found in bars embedded in concrete, which is attributed to the abrasion between the surfaces. An observation of a white residue left on the concrete at the point of friction between the two surfaces was made. Additionally, longitudinal matrix cracks were noticed in the bars before bar rupture occurred.
[114]	V-ROD bar (sand-coated + vinyl-ester) & f′c = 50 MPa	Pull-out (200 × 200 × 200 mm)	No.16 & No.19 bars; 1.5 d_b_, 2.0 d_b_ & 2.5 d_b_ concrete cover	1,000,000 cycles at 1.5 Hz and 25% of GTS	30–50% reduction in bond resistance. Smaller diameter specimens were more affected. The failure mode was mainly pure pullout. Larger concrete cover also impacts the bond strength due to the limitation of microcracks.
[142]	V-ROD bar (sand-coated + modified vinyl-ester) & f′c = 37 MPa	Bridge desk under single concentrated load at mid-span (2500 × 200 × 3000 mm)	Reinforcement ratio (No.16 to No.19)	Accelerated and constant amplitude fatigue loading (min. 15 kN)	The accumulation of damage on the deck slab was observed through an increase in the residual deflection, indicating deterioration. Punching shear was the observed failure mode. Better performance was noticed when compared to that of steel reinforced slabs. The top reinforcement did not influence the behavior.
[143]	Sand-coated + helically wrapped No.16 bar	Bridge slab under two concentrated loads (1500 × 200 × 5000 mm)	Frequency and maximum load in cycles (140 to 440 kN)	Cycles at 0.2 to 1.25 Hz up to failure	The punching failure mode was recorded in all four slab specimens. The maximum static load reduction was approximately 3.4%. All tests proved to have better performance than those of the expected base on the European code.
[144]	Grooved No. 8 bar	Pull-out (200 × 200 × 150 mm)	Concrete strength, concrete cover, and max load in cycles	1,000,000 cycles at 5 Hz and 60/70% of SBS	In the smallest concrete cover (10 mm), fatigue life was constrained by the splitting failure mode. With an increase in the cover, the fatigue failure mode became a mixture of splitting and bar pullout.

**Table 4 materials-16-01990-t004:** Recent studies on creep in GFRP-RC elements.

Study	Materials	Test	Parameters	Condition	Findings
[114]	V-ROD bar (sand-coated + vinyl-ester) & f′c = 50 MPa	Pull-out (5 d_b_ embedment length)	No.16 & No.19 bars; 1.5 d_b_, 2.0 d_b_ & 2.5 d_b_ concrete cover	Sustained (30% of GTS) + Freeze-Thaw Cycles	The bond strength was increased for No. 16 bars with the three concrete covers used. This observation was valid for No. 19 bars with 2.5 d_b_ only. A significant increase in the peak slip (decrease in bond stiffness) was observed for No. 19 bars, while for No. 16 bars, only specimens with 2.5 d_b_ showed an increase in slip.
[113]	9.5 mm sand-coated bar (vinyl-ester) & f′c = 40 MPa	4-point bending test beam (1800 × 130 × 180 mm)	Duration of sustained load (50, 100 & 180 days)	Sustained load at 27% of UTS	Creep strain in the GFRP bars was less than 2.0 % of the initial value after 180 days. The increased rate of deflection was higher during the first 28 days. The ultimate capacity decreased by 3.8%, 4.4% and −6.8%, respectively. However, at the service load limit, the mid-span deflection increased by 2.2%, 1.4% and −1.1%, respectively.
[155]	16 mm GFRP bar & f′c = 35 and 50 MPa	Pull-out test 200 mm side	Duration of sustained load (90 & 130 days) and bond length (5 a 10 d_b_)	Sustained load at 15% of UTS	The results indicate that there was an increase in slip which reached a stable state at about 60 days after loading, regardless of the strength of the concrete. The increments were greater for specimens with longer bond lengths. Additionally, the impact of concrete strength on the stress transfer process and the redistribution of stresses along the bond length was confirmed. As the strength of the concrete increased, there was a decrease in stress levels in the neighboring loaded end zones due to an increase in damage at the bar-concrete interface.

**Table 5 materials-16-01990-t005:** Strength reduction factors for different design guides and codes.

Action	GFRP		Steel	
	ACI 440.11 [15]	AASHTO GFRP [18]	ACI 318-19 [169]	AASHTO LRFD [170]
Moment, axial force, or combined	0.65–0.55 *	0.75–0.55 *	0.65–0.90 *	0.75–0.90 *
Shear	0.75	0.75	0.75	0.85
Torsion	0.75	0.75	0.75	0.85
Bearing	0.65	0.70	0.65	0.70

* Factors varying from the compression- to tension-controlled section.

**Table 6 materials-16-01990-t006:** Environmental reduction factors on FRP reinforced concrete structures.

FRP Composite	ACI 440.11 [15] C_E_	AASHTO GFRP Specifications [18]	CSA S806 [20]	JPCI [17]
Glass	0.85	0.7–0.8	No environmental reduction factor specified	
Carbon	0.9–1.0 *	Not specified		
Basalt	Not specified			
Aramid	0.8–0.9 *			

* Based on ACI 440.1R-15 [13].

**Table 7 materials-16-01990-t007:** Creep rupture reduction factor for tensile strength in different design guides and codes.

	FRP			Steel		
Rebar	ACI 440.11 [15]	AASHTO GFRP [18]	CSA S806 * [20]	Rebar	ACI 318 [169]	AASHTO [170]
GFRP	0.30	0.30	0.25	Steel	No reduction	No reduction
CFRP	0.55 **	-	0.65			
BFRP	Not specified	-	-			
AFRP	0.30 **	-	0.35			

* Multiplied to the guaranteed tensile strength and not the design tensile strength. ** No longer applicable, based on ACI 440.1R-15.

**Table 8 materials-16-01990-t008:** Environmental reduction factors on FRP strengthened concrete structures.

FRP Composite	ACI 440.2R [16]	AASHTO EB Specifications [19]
Glass	0.50–0.75	Not specified
Carbon	0.85–0.95	
Basalt	Not specified	
Aramid	0.70–0.85	

**Table 9 materials-16-01990-t009:** Performance summary of FRP composites under harsh environmental conditions *.

Exposure/Fiber	Glass	Carbon	Observations
Water Exposure *	Excellent	Excellent	Plasticization and hydrolysis causes cracks in the matrix resin. Swelling and dissolution degrade the fiber-matrix bond. Water ingress can also damage the fiber. Overall, fiber delamination and degradation of the FRP bar may cause tensile strength loss and the reduction of both the elastic modulus and the interfacial shear strength.
Saline Exposure *	Good	Good	Diffusion of NaCl or the presence of salts could strain or rupture intermolecular bonds in the matrix resin and the fiber-matrix interface. Composites with a higher fiber volume fraction were found to absorb more seawater compared to ones with lower fiber volume fraction. A seawater solution caused greater moisture absorption than alkaline and tap water solutions. Degradation in tensile strength was observed with a similar behavior as water exposure.
Alkaline Exposure *	Good	Excellent	Alkaline solutions with pH values varying between 11.5 and 13.0 degrade the tensile strength and stiffness of FRP bars. The damage resulting from the presence of alkali is controlled by the matrix at the fiber-resin interface. The interlaminar-shear strength is the most affected property (with moderate degradation), followed by the transverse-shear and tensile strength, respectively.
UV light Exposure	Good	Good	The UV radiation may result in surface oxidation due to different chemical mechanisms related to resin type, which are capable of degrading the molecular bonds in the polymer matrix. The FRP rebars embedded in concrete are themselves protected from UV exposure, but attention should be paid to storage. For external application, UV exposure can cause embrittlement and micro-cracking in an unprotected laminate surface. Color shift or yellowing and gloss changes are some of the indications of UV exposure on FRP laminates.
Elevated Temperature and Fire **	-	-	Due to the differences in the coefficient of thermal expansion, high temperature variations can introduce FRP reinforcement expansion causing the splitting of surrounding concrete. Elevated temperature results in micro cracks at the interface between FRP and the concrete, which decreases the FRP-substrate bond as well as the inter-laminar bond in the FRP.
Freeze-thaw cycles	Good	Good	FRP materials become brittle under freezing temperatures, and absorbed moisture expands when freezing, leading to microcracks, particularly at the interface between the FRP and the substrate material. Exposure to freeze-thaw cycles could affect FRP composites themselves due to mismatched coefficients of thermal expansion of the constituent, resulting in micro-cracking and void generation. However, experimental studies on FRP bars have shown inconsistent behaviors when exposed to freeze-thaw cycles.

* Performance depends on resins type, fiber volume fraction, solution concentration, aging temperature among others. Further, elevated temperature accelerates degradation. ** Elevated temperature due to fire exposure can cause additional degradation. See chapter 3 related to elevated temperature exposure.

## Data Availability

The data presented in this study is available on request from the corresponding author.

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
