# Peer review of "FRP-Reinforced/Strengthened Concrete: State-of-the-Art Review on Durability and Mechanical Effects"

_materials, 2023, doi:10.3390/ma16051990_

Round 1

Reviewer 1 Report

This paper provides a state of the art on the mechanical behavior and durability of fiber-reinforced composites for reinforcement of new concrete members and strengthening of existing concrete members. The paper provides a nice introduction of this technology and covers pretty well the existing literature on the topic. Nevertheless, the discussion of data collected is quite limited and drawing conclusions based on the data provided is not easy. However, the paper remains an interesting reading, which can be recommended for publication after considering the following comments:

1.      In general, the authors focused on (relatively) old contributions found in the literature that, although certainly significant, fail to capture the current trend of composite systems, as also noticed in conclusion #2 (see L810-811). The authors are invited to review the recent literature and improve the discussion provided accordingly (see following comments).

2.      L18: what results of the study? The discussion is quite limited and no new experimental results are provided. I would refer to “findings” or “observations” rather than “results”.

3.      L23: qualitative language should be avoided in scientific papers. What do the authors mean with “high-quality”?

4.      L63-66: the problem of designing concrete structures reinforced with composite bars has been tackled by recent literature on the topic (see e.g., https://doi.org/10.1016/j.conbuildmat.2022.128395).

5.      L108: please specify what “environmental implications” are referenced here.

6.      L111: please use international system units (here and throughout the entire manuscript).

7.      L133: please proof-read the manuscript to amend missing cross references.

8.      Table 1: there is also an Italian guideline to design FRP-reinforced concrete members, namely the CNR-DT 203/2006, which is available for free also in English and that could be considered as well.

9.      L176-183: the authors should better explain why the industry is currently focusing on GFRP bars rather than other types of bar.

10.  Figures 2, 3, and 4: analogous and more deep analyses as that shown in these figures were reported in https://doi.org/10.1016/j.compositesb.2017.12.037 and https://doi.org/10.1016/j.conbuildmat.2018.07.175. This last paper is also useful to clarify the role of the applied stress on the durability of GFRP bars, mentioned in L296-298.

11.  Line 231-247: also in this case the authors referred only (relatively) to old literature, while more recent contributions can be found (see e.g., https://doi.org/10.1016/j.compstruct.2020.111947).

12.  L302: the transverse shear strength does not seem directly related to the behavior of stirrups. This point should be better clarified.

13.  L350-351: please use the term “bar” instead of “rod” for consistency with the rest of the manuscript.

14.  Section 3.5: a thorough state of the art on these phenomena has been recently published in https://doi.org/10.1016/j.compstruct.2022.116273.

15.  L504-516: a state of the art on the fatigue behavior of bare and embedded GFRP bars was reported in https://doi.org/10.1016/j.conbuildmat.2022.128395.

16.  Section 4.2: also in the case of creep behavior of composite bars, the authors should revise the recent literature (e.g., https://doi.org/10.1016/j.compstruct.2019.111283).

17.  L621-625: this description is too general and not accurate. Please revise.

18.  L745-747: this statement is questionable. The contribution of the externally bonded composite should be accounted for also during the elastic behavior of steel. Please note that in shear strengthening of RC members there is a phenomenon, referred to as steel-composite adverse interaction, for which failure of strengthened beams may occur before yielding of the steel reinforcement.

19.  L797: the discussion on thermoplastic resins is quite limited. Again, the authors should revise the recent literature on the topic (e.g., https://doi.org/10.1016/j.conbuildmat.2022.130104, https://doi.org/10.3389/fmats.2019.00290).

L826: please specify what documents are referenced here.

Reviewer 2 Report

As an overview paper, this paper systematically summarizes the mechanical performance and long-term durability of FRP-Reinforced/Strengthened concrete structures. Furthermore, some provisions for the serviceability design of FRP-RSC elements are commented for the readers to understand the implications of the durability and mechanical properties. Overall, the paper is well designed with rich information. The following comments should be further replied to make necessary improvements.

1.      Title

(1) FRP is suggested to use the full name when occurring in the title.

2. Abstract

(1) Please add some quantitative results and conclusions on the properties of FRPs and serviceability design method of FRP-RSC elements through current review and literature review.

(2) The application of FRP reinforcement and strengthening should be further clarified through the GFRP bars and CFRP fabrics owing to the different service environments. 

3. Introduction

(1) The authors are suggested to make the summary on the main concern according to the different applications of FRP reinforcement and strengthening of RC structures. For the reinforcement application (such as GFRP), The long-term performance degradation of GFRP bars under the alkaline environment and loading is the main problem. For the strengthen application (such as CFRP), the reliable anchoring performance of CFRP (prestressed strengthen) and the debonding of the interface between CFRP and concrete should be considered.

(2) The long-term performance degradation mechanism (such as fatigue damage mechanism, creep mechanism) of FRPs and FRP-concrete interface exposed to the environmental and loading conditions should be briefly summarized, for example, resin matrix cracking, fiber etching, interface debonding fiber/resin, FRP concrete interface peeling and so on.

(3) The resin is a typical polymer material, so it is sensitive to high temperature, which also leads to the obvious performance degradation of fiber reinforced polymer composite at high temperature. As the harsh environmental conditions, high temperature is a typical environmental factor, which should be considered and further summarized.

The authors are suggested to consider the above comments to further enrich the summary work of the current introduction by reviewing the latest typical research below. Polymer Composite, 2020, 41:5143-5155. Composite Structures. 2021, 261: 113285.  Composite Structures, 2022, 293, 115719. 

4. Background on FRP Composites

(1) For the internal application, the authors stated: FRP’s initial cost is also frequently highlighted as one the main drawbacks when compared to the price of steel rebars,”, however, from the perspective of the Life Cycle Assessment (LCA), the overall cost of FRP reinforced concrete structures is lower than that of steel reinforced concrete structures. It is suggested that the authors carry out relevant analysis from this perspective.

(2) For most common externally bonded FRP systems for strengthening existing structures, prestressed technology is an effective reinforcement method. In this system, the design and performance of anchorage system is critical to ensure the high performance and long-term reliability of the strengthening system. Relevant analysis and summary should be considered. 

5. Environmental effects

(1) For the 3.1 and 3.2, the effects of water exposure and saline exposure on the long-term performance of FRPs should be have no significant difference because there is no relevant chemical reaction. The main degradation of FRPs is caused by the diffusion of water molecules. It is suggested that the authors make a simple comparison and draw some relevant conclusions at the end of section 3.2.

(2) For the alkaline exposure, the reaction mechanism between glass fiber and concrete alkali solution should be further mentioned through mechanism diagrams or chemical reaction equations.

(3) For UV exposure, the relevant application background should be added, because when FRP will not be completely exposed to the environment when some radiation resistant coatings or casing are considered. 

6. Mechanical effects

(1) The summary of fatigue and creep damage mechanism of FRPs should be provided for the readers. 

7. Conclusions

The conclusion is suggested to be further condensed to 4~5 key information points. The rest can be put in the future prospects. 

Reviewer 3 Report

There are some minor points here that need to be addressed.

Page 3, line 133: “(see Error! Reference source not found.)”. Fix this.

Also page 4, line 168: “(see Error! Reference source not found.)”

Section “3.2. Saline exposure”: There are more recently published works regarding the effect of seawater/ salt water on the mechanical properties of GFRPs and CFRPs on macro-scale and micro-scale such as “Indentation characterization of glass/epoxy and carbon/epoxy composite samples aged in artificial salt water at elevated temperature”, and “Hygrothermal deterioration in carbon/epoxy and glass/epoxy composite laminates aged in marine-based environment (degradation mechanism, mechanical and physicochemical properties)”. And, also an MDPI (journal of materials) recently published paper regarding the Thermally Damaged Concrete under a Hygrothermal Environment can be added to the section “3.6. Elevated temperature exposure”. “Experimental Study of Thermally Damaged Concrete under a Hygrothermal Environment by Using a Combined Infrared Thermal Imaging and Ultrasonic Pulse Velocity Method”

Explain (and add) the difference between the effect of a saline environment and a water environment on the mechanical properties of composite materials.

The focus of the review paper is on the effect of environmental factors on the mechanical properties of thermoset composites, so highlight/emphasize this in the title, abstract and last paragraph of the introduction section.

The composite materials reinforced with glass fiber and carbon fiber are discussed in this paper. What about the other types of fibers?

Incorporate the summary section and conclusion section in a single section.

Round 2

Reviewer 1 Report

The authors only addressed some of the comments provided, giving evasive if none responses for others. Comments provided were meant to improve the quality and clarity of the paper. Authors may decide not to consider them providing convincing reasons for rebuttals. The authors are invited to seriously account for the comments provided: 

1.     The new references added were neither mentioned nor indicated/marked in any way in the revised manuscript.

3.     “outstanding properties” is still qualitative and not sufficiently clear for a Journal paper. Outstanding with respect to what? 

4.     The authors did not understand the general comment regarding the limited presence of recent works from the literature. As they mentioned, the technology has been improving and relatively old results might not be representative of current trends. This does not mean that previous works should be disregarded, yet recent contributions should be better referenced as they represent the basis for revising current code/guidelines. It is not clear what new reference was added in the revised manuscript (see comment #1).

5.     This response is still not clear. Details should be added to explain what the authors mean with “sustainability implications”, which are not obvious. Also, no revision was implemented in the manuscript, which still reports “environmental implications” (L117).

7.     An error is still present in L180. Again, carefully proof-read the manuscript before submission. 

9.     Something is missing in the sentence in L180-193, which does not make sense.  

10.  The authors disregarded the first part of the comment. Figures similar to Figures 2, 3, and 4 could be found in the literature, as also confirmed by the response provided. Although this is a review paper, works that made the same analysis before the authors should be properly acknowledged. 

15.  The same remarks made to the previous comment (#10) hold here. Previous research should be properly acknowledged in a review paper. Since the analysis reported here and in Figure 2, 3, and 4 are limited with respect to previous analyses that can be found in the literature, a reader would appreciate references to other works with deeper analyses of the topic with respect to those found in this manuscript. 

19.  This response is not convincing. Only the main findings of the paper shall be reported in the conclusions. If the authors does not consider research on thermoplastic resins valuable, then this type of resin should not be mentioned in the conclusions. 

20.  The entire name (code) of AASHTO and ACI documents shall be reported for clarity.  

Reviewer 2 Report

Thank you for your efforts to revise the manuscript, and the authors have replied to most of the comments. Two small comments can be considered as following:

1. For the points 4~7, the authors have made a detailed summary of the content of each part, which is good. However, in the first part of introduction, a brief summary and analysis should be provided for the readers to understand the information. As a review paper, the summary of the introduction should be comprehensive and systematical, including the content of each part. Then the content of each part should be detail summarized based on the information in the introduction.

2. Please indicate the revised information in the text on point 13 "The summary of fatigue and creep damage mechanism of FRPs should be provided for the readers."
